# Co-Occurring Depression and Associated Healthcare Utilization and Expenditure in Individuals with Respiratory Condition: A Population-Based Study

**DOI:** 10.3390/pharmacy9040157

**Published:** 2021-09-25

**Authors:** Prashant Sakharkar, Thanh Mai

**Affiliations:** 1Department of Clinical and Administrative Sciences, College of Science, Health and Pharmacy, Roosevelt University, Schaumburg, IL 60173, USA; 2College of Pharmacy and Health Sciences, Western New England University, Springfield, MA 01119, USA; thanh.mai@wne.edu

**Keywords:** depression, asthma, COPD, chronic obstructive pulmonary disease, health expenditure, cost of care, health services utilization, healthcare utilization

## Abstract

The existing literature is limited on the prevalence of depression among people with respiratory conditions and person-level factors that are associated with increased healthcare utilization and expenditures. The aim of this study was to explore the prevalence, pattern of healthcare use, and expenditures in noninstitutionalized individuals having co-occurring depression with respiratory conditions. The Medical Expenditure Panel Survey (MEPS) data from 2011 to 2017 was used in this study. Our sample included individuals having respiratory conditions (asthma, emphysema, and chronic bronchitis) with and without depression. Healthcare use and expenditure data were analyzed using a chi-square test, *t*-tests, and multiple linear regression analyses. There were 8848 individuals in the study. The prevalence of comorbid depression was 20%. Individuals with co-occurring depression with respiratory conditions differed significantly from individuals without co-occurring depression for age ≥ 45 years, white, and with ≤2 chronic disease conditions. Depressed individuals with respiratory conditions had higher healthcare utilization and expenditures. The presence of co-occurring depression with respiratory conditions increases the treatment complexity, healthcare utilization, and expenditure. Better treatment and management of these patients may reduce healthcare use and expenditures in the future.

## 1. Introduction

Respiratory conditions such as asthma and chronic obstructive pulmonary disease (COPD) are associated with high costs and economic burdens among the US population. This cost was estimated with annualized total medical expenditures being USD 7 billion for asthma and USD 5 billion for COPD in 2017 [1,2,3,4]. COPD, which includes emphysema and chronic bronchitis, is the fourth leading cause of death in the US [5]. It has been shown to have a higher prevalence in individuals with lower income, women, living in the rural areas, having public insurance, and a history of smoking [6,7]. The prevalence of asthma is also increasing, with approximately 8% (25.7 million people) in the US having been diagnosed with or have had asthma [4].

Both asthma and COPD have shown higher rates of comorbidity mostly associated with mental health disorders such as anxiety and depression [8]. The prevalence of depression in COPD appears to be more frequent, compared to other chronic disabling diseases. This may be due to the increased disability and impaired quality of life associated with the disease. Asthma is also linked with mental health disorders, particularly depression [9,10]. This may be due to high stress caused by the financial and physical hardships associated with the condition and treatment. Several reports have suggested that those with asthma have higher rates of anxiety and depression, compared to patients with other chronic diseases such as chronic hepatitis [9,10]. 

Although depression is common in both asthma and COPD, it seems that co-occurring depression with asthma and COPD is either left untreated or inadequately treated. There are several treatment options for depression in COPD and asthmatic patients, with antidepressants being one of the most often used. The use of antidepressants helps in reducing the symptoms of depression. However, their use may be associated with a decline in quality of life resulting from sexual dysfunction, feeling emotionally numb, addiction, further exacerbations of disease conditions with withdrawal, and even death due to overdose or toxicity [11]. Deb et al. explored the rates of depression with antidepressant use and combination therapy, treatment of depression by demographic and socioeconomic status, access to care, health status, and personal healthcare practices among adults. The findings of this study concluded that the treatment of depression among adults with COPD needs to be tailored to different subgroups [12]. 

In a study on the effects of depression on healthcare expenditures among older individuals, people who are depressed were found to spend far more for healthcare, compared to their nondepressed counterparts [13]. Similarly, among managed care population in the United States, COPD patients with co-occurring depression were found 77% more likely to be hospitalized, 48% more likely to have an emergency room (ER) visit, and 60% more likely to be hospitalized/visit the ER, compared to COPD patients without co-occurring depression [14]. 

To date, the available literature is still limited on the prevalence of depression among people with respiratory conditions. Person-level factors in individuals with co-occurring depression with asthma or COPD, and patterns and predictors of healthcare use and expenditure have been rarely explored. 

The aim of this study was to explore the prevalence, pattern of healthcare use, and expenditures in noninstitutionalized individuals having co-occurring depression with respiratory conditions.

## 2. Materials and Methods

In this retrospective study, US Medical Expenditure Panel Survey (MEPS) data from 2011 to 2017 was used to determine the prevalence, healthcare use, and expenditures associated with respiratory conditions with and without co-occurring depression. Multiple years’ survey data were used to achieve an adequate sample size [14]. MEPS is a nationwide survey of the US population and their healthcare providers. Employers through this survey provide nationally representative estimates of healthcare use, expenditures, and insurance coverage for the civilian noninstitutionalized population [15]. The MEPS collects data from a sample through an overlapping panel design. A new panel of sample households is selected each year, and data for each panel are collected for two calendar years in five rounds of interviews that occur over a two-and-a-half-year period. This provides continuous and current estimates of healthcare expenditures at both the person and household levels for two panels, for each calendar year [15]. In this investigation, we consolidated data for the years 2011 to 2017 to identify demographic information, total healthcare utilization, and expenditures over this length of time between depressed and nondepressed individuals with respiratory conditions.

MEPS is cosponsored by the Agency for Healthcare Research and Quality (AHRQ) and the National Center for Health Statistics (NCHS). There are three components in the MEPS survey. The household component collects person-level data on demographic and socioeconomic characteristics, health conditions, use of medical services, charges and payments, access to care, employment, health insurance coverage, and income. The medical provider component acts as a supplement and validates the household survey; a sample of medical providers are contacted to obtain information that household respondents could not accurately provide. In addition to collecting data on all medical and pharmacy events at the person level, information is also collected on dates of visits, diagnosis and procedure codes, and charges and payments. Diagnoses in MEPS are reported according to International Classification of Diseases-9-Clinical Modification (ICD-9-CM) codes and ICD-10 codes (ICD-10-CM); office-based visits are reported according to Current Procedural Terminology, 4th Edition (CPT-4) codes; and prescription names, strength, and quantity dispensed are also collected as part of the medical provider component. The insurance component is conducted by the US Census Bureau and collects information on establishment characteristics such as whether health insurance is offered and details on health plans. ICD-9 and ICD-10 codes and clinical classification codes are converted by professional coders. The MEPS survey panel design includes 5 rounds of interviews covering 2 full calendar years.

Each self-reported medical condition is verified by contacting medical providers and pharmacies that the respondents identified as their source of care. The household respondents’ medical conditions, therefore, typically correlate with medical care provider data; MEPS participants’ reported conditions have been demonstrated to be consistent with those provided by their medical providers for chronic conditions (e.g., diabetes, hypertension, mental health, etc.) with a median sensitivity of 70% [16].

The MEPS medical conditions file was used to identify medical conditions, the prescription use file was used to identify medication use, and the full-year consolidated data were used to identify demographic information and expenditures related to conditions of interest.

### 2.1. Study Population

The study population consisted of individuals who had respiratory conditions (i.e., asthma and COPD broken down into chronic bronchitis and emphysema) with and without depression and who were on antidepressant medications. The respiratory conditions were identified using ICD-9 cm code 491 (asthma), 492 (chronic bronchitis), and 493 (emphysema), and 311 (depression) for the years 2011 to 2014. ICD-10 cm codes from the years 2015 to 2017 were converted to ICD-9 cm codes using a crosswalk.

Person-level variables that were analyzed included age (<25, 25–44, 45–64, 65 yrs. and above), sex (female; male), race/ethnicity (White, Asian, Black, Hispanic), marital status (married, not married/other), education level (less than high school, high school/GED, associate/bachelors, and higher than bachelor’s degree), comorbid chronic disease conditions (2 or less, 3–5, greater than 5), health insurance status (private, public, and uninsured), and region (Northeast, Midwest, South, and West). The incomes of respondents were reported as categories based upon percentages of the federal poverty level. The five categories of income status included negative or poor (less than 100%), near poor (100% to less than 125%), low income (125% to less than 200%), middle income (200% to less than 400%), and high income (greater than or equal to 400%). The quality of life (QOL) indicators included limitations of activities of daily living (ADLs; yes/no), limitations of instrumental activities of daily living (IADLs; yes/no), and pain (quite/extreme, little/moderate, no pain). We treated missingness at random for continuous variables and used an imputation approach by adding an extra category for the categorical variables.

### 2.2. Chronic Disease Conditions

Certain chronic disease conditions in MEPS data were identified beforehand as priority conditions due to their prevalence, costs, and/or based on previously well-accepted standards of clinical care for such conditions. Major long-term and life-threatening conditions such as cancer, high cholesterol, hypertension, ischemic heart disease, and stroke are included in MEPS and were also included in this study. An individual was then identified as having a comorbid chronic disease condition if any one of these seven chronic life-threatening conditions was present in addition to having respiratory conditions and/or depression. These conditions were categorized into three groups: ≤2, 3–5, and >5 chronic disease conditions.

### 2.3. Healthcare Utilization

Four categories of healthcare utilization were analyzed, which included ambulatory visits, emergency department visits, hospital inpatient days, and prescription medication use. An ambulatory visit is defined as a medical provider visit in an office-based setting, hospital outpatient department, or hospital admission with a zero-night stay (i.e., admission and discharge occurred on the same day). Medical provider visits were defined as visits to physicians/nonphysicians (e.g., physician assistants, nurse practitioners, physical and occupational therapists, social workers, etc.). Other nonphysician visits were defined as visits to nurses, optometrists, psychologists, and technicians/other medical providers. Hospital outpatient department visits included visits to physician and nonphysician providers. An emergency department visit was defined as all visits made to the emergency department which included visits that ended in an inpatient stay. A hospital inpatient stay was defined as a hospital admission that resulted in the patient staying at least one night before discharge for that hospitalization. Prescription medication use was defined as days of prescribed medications supply in the years 2011–2017. Data were log transformed. and the means of the log-transformed values were used for determining statistical significance. 

### 2.4. Healthcare Expenditure

MEPS defines expenditures as the sum of direct payments for care provided during the year. Expenditures included out-of-pocket (OOP) payments, payments made by third parties (Medicare, Medicaid, private insurance), and payments made from other sources. Payments made for over-the-counter medications, payments for alternative care services, and indirect payments not related to specific medical events were not included as an expenditure. In summary, the total amount paid for health services as an OOP cost and by third-party payers between 2011 and 2017 represented the total healthcare expenditure. 

Ambulatory expenditures included payments made for office-based provider visits, hospital outpatient visits, and zero-night hospital stays. Both facility-based and direct provider expenses were included in payments for hospital outpatient visits and zero-night stays. Emergency department expenditures included both facility and direct provider expenses for emergency department visits; expenditures for hospital stays also included facility and direct expenses. Expenditures for prescription medications included expenses for purchased medications only and excluded any expenses associated with sample medications so as to reflect OOP and third-party payments. Other categories of use included vision aids, other medical equipment, and services (e.g., ambulance services, orthopedic items, prostheses, bathroom aids, and disposable supplies). 

The expenditure data were also log transformed, and the means of the log-transformed values were used for determining statistical significance. Mean expenditures were adjusted to the 2020-dollar value by using the Medical Consumer Price Index [17].

### 2.5. Statistical Analysis

Data were analyzed for descriptive statistics and associations using SPSS ver. 27 (IBM, Armonk, NY) and STATA ver. 14 (StataCorp LP, College Station, TX) to account for the MEPS complex sampling design and to incorporate sample weights. An independent *t*-test was used to compare healthcare utilization and expenditures in individuals with depressed vs. nondepressed with respiratory conditions, whereas the chi-square test was used to compare demographic characteristics in those with depressed and nondepressed individuals with respiratory conditions.

A multiple linear regression analysis was performed to identify the predictors of the healthcare utilization and total healthcare expenditures adjusting for age, sex, race/ethnicity, co-occurring depression, health insurance, comorbidity, and region. These covariates were chosen based on the prior evidence in the literature. Log-transformed values of total healthcare utilization and expenditure were used as dependent variables. A *p*-value of ≤0.05 was considered statistically significant.

## 3. Results

### 3.1. Prevalence of Depression

Our study sample included 8848 individuals with respiratory conditions, of which 1792 (weighted count of 18,622,014 ≈ 18.6 million) also had depression. The prevalence of depression among individuals with respiratory conditions in our sample was 20%, suggesting one in five individuals with respiratory conditions also suffered from depression. 

### 3.2. Demographic and Clinical Characteristics

Overall, greater proportion of individuals over 45 years of age had depression with respiratory conditions (88% vs. 36%; *p* < 0.001); were white (82.6% vs 0.7%; *p* < 0.001); with high school/General Educational Development (GED) education (61.4% vs. 8.3%; *p* < 0.001); not married (54% vs. 46%; *p* < 0.001); with two or less chronic disease conditions (45.3% vs. 9.6%; *p* < 0.001); had middle income (29% vs. 8.1%; *p* < 0.001); with private insurance (49.6% vs. 4.4%; *p* < 0.001); in quite or extreme pain (40.2% vs. 20%; *p* < 0.001); with ADL disability (11.2% vs. 2.9%; *p* < 0.001); IADL disability (17.2% vs. 5%; *p* < 0.001) and living in southern region of the US (42.8% vs. 34.4%; *p* < 0.001) (Table 1). 

### 3.3. Healthcare Utilization 

Depressed individuals with respiratory conditions had more ambulatory care visits than their nondepressed counterparts (24.5 vs. 17.7; *p* < 0.001). Similarly, depressed individuals with respiratory conditions filled more prescriptions (51.6 vs. 24.3; *p* < 0.001), had more days of prescription medication supply (23.7 vs. 17.1; *p* < 0.001), had more emergency department visits (0.75 vs. 0.38; *p* < 0.001), and hospital inpatient days (2.75 vs. 0.77; *p* < 0.001), compared to nondepressed individuals with respiratory conditions (Table 2). 

### 3.4. Healthcare Expenditure

The expenditures for ambulatory visits were higher in depressed, compared to nondepressed individuals with respiratory conditions (USD 16,596 vs. USD 9377, *p* < 0.001). Similarly, emergency department (USD 836 vs. USD 442; *p* < 0.001), hospital inpatient (USD 7931 vs. USD 2836; *p* < 0.001), prescription medication (USD 8187 vs. USD 3737; *p* < 0.001), and other medical expenditures (USD 359 vs. USD 227; *p* < 0.001) were also higher in depressed than nondepressed individuals with respiratory condition. The total healthcare expenditures of individuals with co-occurring depression with respiratory conditions were more than their nondepressed counterparts (USD 24,532 vs. USD 12,420; *p* < 0.001). This suggests that total healthcare expenditures almost doubled if individuals had co-occurring depression with respiratory conditions (Table 3).

### 3.5. Predictors of Healthcare Utilization and Expenditure

The full regression model with seven covariates explained 12.1% of the variance (F(7312) = 81.01; *p* < 0.001) (Table 4). Independent variables within the model contributed significantly to the model except for co-occurring depression with respiratory condition and region; age (B = 0.24; *p* < 0.001), sex (B = 0.33; *p* < 0.001), race/ethnicity (B = 0.09; *p* < 0.001), insurance (B = −0.07; *p* < 0.05), and chronic disease conditions (B = 0.31; *p* < 0.001) (Table 4). Those who had more than two chronic disease conditions, being female, and other than white race/ethnicity were more likely to have increased healthcare utilization by 31%, 33%, and 9%%, respectively, respectively. On the other hand, having insurance had a counterintuitive decrease in healthcare utilization by 7%. In terms of healthcare expenditures, regression model explained 24% of the variance (F (7312) = 182.6; *p* < 0.001). Independent variables included in the model contributed significantly to the model; age (B = 0.19; *p* < 0.001), sex (B = 0.14; *p* < 0.001), co-occurring depression with respiratory conditions (B = 0.10; *p* < 0.001), chronic disease conditions (B = 0.29; *p* < 0.001), insurance (B = −0.06; *p* < 0.001), and region (B = −0.02; *p* < 0.05) (Table 4) were found to be a good predictor of healthcare expenditure. Increasing age, having a high number of chronic disease conditions, and being female were found more likely to increase total healthcare expenditure by 19%, 29% and 14%, respectively, whereas having insurance and living in other than the West were found more likely to decrease the total healthcare expenditures by 6% and 2% among individuals with co-occurring depression with respiratory conditions (Table 4).

## 4. Discussion

This study estimated the national prevalence of co-occurring depression among community-dwelling individuals with respiratory conditions in the US. The varying level of prevalence of depression among individuals with respiratory conditions, with asthma and COPD was reported independently in the existing literature. COPD with concomitant depression had an estimated prevalence ranging from 10% to 57%, which can be attributed to the different measurement tools and different degrees of illness severity across the studies [18]. Similarly, the prevalence of depression in asthma has ranged from 1% to 45%, which is also likely due to different sampling techniques that could have led to the wide variations in the reported prevalence [19]. Although it is hard to draw conclusions based on these discrepancies, most of these findings suggest that there is an increased prevalence of depression in patients with asthma and COPD. Our findings are well within the reported ranges.

Significant differences were observed for the demographic characteristics among depressed and nondepressed individuals with respiratory conditions in this study. Depressed individuals with respiratory conditions were more likely to be older, of the female gender, of white ethnicity, high school graduate, unmarried, and within a middle-income group. They were also more likely to have two or fewer comorbid conditions, have private insurance, living in the South, having quite/extreme pain, and with ADL and IADL disabilities. In one study, individuals who were female, African American, had low household income, with high school or less education level, and lacked private insurance had higher rates/severity of depression [20]. Participants in this study were mostly between the ages of 18 and 30 and the majority from white and Black ethnicities, whereas, in this study, we did not restrict participants by age or ethnicity. These differences in the results could be attributed to the differences in inclusion criteria. We also used different threshold values for creating categories of poverty status and number of chronic disease conditions. 

Earlier studies have reported increased healthcare utilization and expenditures associated with co-occurring depression in the general population [21,22,23], as well as increased costs and utilization in individuals with asthma and COPD [1,24,25]. In this study, healthcare utilization was categorized into four categories so that the effects of healthcare utilization can be ascertained for each category independently rather than aggregating healthcare utilization as a total, which was commonly reported in other studies [14,26,27]. A study on asthma/COPD overlap syndrome reported an increase in ER visits and hospitalizations only [25], whereas, in this study, ambulatory visits and prescription medications use were major areas of differences between depressed and nondepressed individuals with respiratory conditions, in addition to emergency department visits and hospitalization. In a cohort of patients with anxiety/depression with COPD in a managed care population, 77% of patients were found more likely to have COPD-related hospitalizations, 48% more likely to have emergency department visits, and 60% more likely to be hospitalized [20]. Similar trends were seen in a study looking at comorbid depressive disorders and asthma where an increase in primary care visits, emergency department visits, other outpatient visits, and more pharmacy fills were reported [26], although this study was conducted among the adolescent population. A cross-sectional survey of patients with COPD in primary care clinics concluded that depressive symptoms are associated with an increase in healthcare utilization, particularly in physician visits, ER visits, and hospitalizations for lung disease [27]. Our results endorsed these earlier findings but also further validated the relationships between depression and respiratory conditions and its impact leading to increased healthcare utilization. 

Healthcare expenditures in all six categories (ambulatory, emergency department, hospital inpatient, prescription medication, other medical, and total expenditures) were higher for the depressed individuals, compared to nondepressed individuals with respiratory conditions. There has not been any current literature exploring expenditures in these categories of depressed and nondepressed individuals with respiratory conditions. The total expenditures for depressed individuals were estimated to be over USD 24,000, twice as much as compared to the nondepressed individual group. Studies have concluded that COPD with co-occurring conditions increases the total annual cost, hospitalization, and ER visits: ER visits are the largest contributor of this cost, while in the case of asthma, these costs increase as much as three times per capita [19]. When the cost of illness related to asthma was assessed in one study, the inpatient hospital services and ER visits contributed the most to the total direct costs [28]. In another study, the total healthcare costs for comorbid depression and asthma increased by 51%, compared to individuals without depression among adolescents [26]. Our results indicated that the largest contributors to the total expenditures were ambulatory, inpatient, and prescription medications; the difference in the ambulatory expenditures could possibly be explained by the fact that we included all physician and nonphysician visits. 

Individuals living in the West accounted for the largest spending in healthcare, which would likely be due to the factors such as type of population, healthcare access and coverage, and lifestyle [27]. The cost estimates for total healthcare expenditures for respiratory conditions with depression differed from previous economic studies, most likely due to differences in perspectives and assumptions used in these studies for estimating healthcare utilization and expenditures. 

There are several limitations that need to be considered when interpreting the results of our study. First, individuals included in the study sample represent the noninstitutionalized population living with depression and respiratory conditions and seeking treatment. The prevalence of depression and respiratory conditions reflects treated prevalence based on reported healthcare utilization. Thus, the prevalence estimates observed in this study could be lower than and underestimate the true disease prevalence because of underreporting and misclassification bias. Second, the healthcare use and expenditures were estimated based on two-year healthcare utilization and expenditure for an individual and represented a snapshot of a short-term effect of depression in individuals with respiratory conditions. Since this is a retrospective study, we cannot ascertain a temporal relationship. We also could not conclude that increased healthcare utilization and expenditures observed in this study among depressed individuals with respiratory conditions are solely due to depression or combinations of one or more of the respiratory conditions (asthma, chronic bronchitis, or emphysema), or other clinical characteristics or behavioral factors. Moreover, we could not assess the severity of depression or respiratory conditions due to the lack of laboratory or diagnostic data in the MEPS. MEPS data are not primarily designed to facilitate state or local estimation of healthcare utilization or expenditure; hence, such analyses could not be undertaken. Similarly, imputation for missing data may have influenced our results. Finally, our study only measured direct healthcare expenditures associated with co-occurring depression with respiratory conditions. We did not assess the indirect costs such as costs associated with loss of productivity and diminished quality of life in this study. 

## 5. Conclusions

The presence of depression increases the complexity of treatment in individuals with respiratory conditions. This further leads to increase healthcare utilization and expenditure where certain subpopulations are at more risk than the others. Aggressive treatment and follow-up for individuals with depression with respiratory conditions may provide greater benefits to these patients and are likely to reduce healthcare utilization and expenditure in the future. 

## Figures and Tables

**Table 1 pharmacy-09-00157-t001:** Demographic characteristics of depressed and nondepressed individuals with respiratory conditions.

	All N (%)	Nondepressed n (%)	Depressed n (%)	*p*-Value
	8848	7056	1792	
**Age (Years)**				
<25	3304 (37.3)	3282 (46.5)	22 (1.2)	<0.001
25–44	1411 (15.9)	1218 (17.3)	193 (10.8)	
45–64	2230 (28.6)	1740 (24.7)	790 (44.1)	
≥65	1603 (18.1)	816 (11.6)	787 (43.9)	
**Sex**				
Male	3672 (40.7)	2990 (40.7)	682 (40.9)	0.899
Female	5176 (59.3)	4066 (59.3)	1110 (59.1)	
**Race/Ethnicity**				
White	3944 (68.5)	2795 (64.4)	1149 (82.6)	<0.001
Asian	308 (2.5)	282 (3.0)	26 (0.7)	
Black	2128 (14.9)	1811 (16.4)	317 (9.8)	
Hispanic	2044 (14.1)	1819 (16.2)	225 (6.9)	
**Education**				
<High school	1358 (21.7)	1239 (25.8)	119 (8.3)	<0.001
High school/GED	2088 (42.5)	1423 (36.6)	665 (61.4)	
Associate/bachelor’s degree	1058 (25.9)	804 (26.3)	254 (24.8)	
>Bachelor degree	348 (9.9)	299 (11.3)	49 (5.5)	
**Marital Status**				
Married	2496 (35.3)	1786 (32.2)	710 (46.0)	<0.001
Not married/Other	6352 (64.7)	5270 (67.8)	1082 (54.0)	
**Comorbid Disease Groups**				
≤2	6806 (74.9)	5973 (83.4)	833 (45.3)	<0.001
3–5	1781 (21.8)	985 (15.1)	796 (45.1)	
>5	261 (3.3)	98 (1.5)	163 (9.6)	
**Poverty Level** ^a^				
Poor	2585 (19.3)	2083 (18.6)	502 (21.5)	<0.001
Near poor	656 (5.8)	488 (5.2)	168 (8.1)	
Low income	1398 (14.2)	1083 (13.4)	315 (17.1)	
Middle income	2175 (26.5)	1703 (25.8)	472 (29.0)	
High income	2034 (34.2)	1699 (37.0)	335 (24.3)	
**Insurance**				
Private	4202 (60.4)	3457 (63.5)	745 (49.6)	<0.001
Public	4272 (35.9)	3317 (33.0)	955 (46.0)	
Uninsured	374 (3.7)	282 (3.5)	92 (4.4)	
**Pain**				
Quite/Extreme	1610 (26.4)	920 (20.5)	690 (40.2)	<0.001
Little/Moderate	2130 (40.5)	1493 (40.8)	637 (39.8)	
No pain	1678 (33.1)	1390 (38.7)	288 (20.0)	
**ADL Disability**				
Yes	442 (4.8)	217 (2.9)	225 (11.2)	<0.001
No	8294 (95.0)	6742 (96.8)	1552 (88.7)	
**IADL Disability**				
Yes	698 (7.8)	364 (5.0)	334 (17.2)	<0.001
No	8041 (92.0)	6599 (94.7)	1442 (82.6)	
**Region** ^b^				
Northeast	1701 (19.4)	1434 (20.2)	273 (16.6)	<0.001
Midwest	1937 (23.1)	1506 (22.9)	431 (23.7)	
South	3147 (36.1)	2415 (34.4)	759 (42.8)	
West	1971 (20.4)	1671 (21.9)	300 (15.2)	

^a^ Under poverty level, poor were defined as with income <100% of the federal poverty line (FPL); near-poor income as 100–<125% of FPL; low-middle income as 125–<200% of FPL; middle income as 200–<400% of FPL; high income as ≥400% of FPL; ^b^ Under region, Northeast included following states (CT, ME, MA, NH, NJ, NY, PA, RI, VT); Midwest (IN, IL, IA, KS, MI, MN, MO, NE, ND, OH, SD, WI); South (AL, AR, DE, DC, FL, GA, KY, LA, MD, MS, NC, OK, SC, TN, TX, VA, WV); West (AK, AZ, CA, CO, HI, ID, MT, NV, NM, OR, UT, WA, WY). GED: General Educational Development. ADL: activities of daily living. IADL: instrumental activities of daily living.

**Table 2 pharmacy-09-00157-t002:** Comparison of mean healthcare utilization among depressed and nondepressed individuals with respiratory conditions.

	Nondepressed			Depressed	
Utilization Category	n	Mean Utilization (SE)	n	Mean Utilization (SE)	*p*-Value ^a^
Ambulatory visits	6373	17.7 (0.43)	1729	24.5 (0.75)	<0.001
Emergency department visits	1916	0.38 (0.01)	681	0.75 (0.03)	<0.001
Hospital inpatient days	746	0.77 (0.06)	488	2.75 (0.20)	<0.001
Prescribed medication use (days of supply)	7053	17.1 (0.48)	1792	23.7 (0.93)	<0.001
Number of prescription medications	7056	24.3 (0.59)	1792	51.6 (1.21)	<0.001

^a^ *p*-value for mean log-transformed utilization. SE: standard error.

**Table 3 pharmacy-09-00157-t003:** Comparison of mean healthcare expenditures among depressed and nondepressed individuals with respiratory conditions.

		Nondepressed		Depressed	
Expenditure Categories	n	Mean Expenditures ^a^ (USD) (SE)	n	Mean Expenditures ^a^ (USD) (SE)	*p*-Value ^b^
Ambulatory expenditures	6368	9377 (581.2)	1722	16,596 (1032.9)	<0.001
Emergency department expenditures	767	442 (28.3)	653	836 (86.2)	<0.001
Hospital inpatient expenditures	746	2836 (344.9)	488	7931 (901.9)	<0.001
Prescription medication expenditures	7056	3737 (152.8)	1791	8187 (672.8)	<0.001
Other medical expenditures	1627	227 (37.2)	596	359 (38.1)	<0.001
Total expenditures	7056	12,420 (514.2)	1792	24,532 (1275.5)	<0.001

^a^ Expenditures are adjusted for inflation with the Medical Consumer Price Index to reflect 2020 dollars. ^b^ *p*-value for mean log-transformed expenditures.SE: standard error.

**Table 4 pharmacy-09-00157-t004:** Co-occurring depression with respiratory conditions in terms of healthcare utilization and healthcare expenditures on regression analysis.

	Healthcare Utilization ^a^		Healthcare Expenditure ^a^	
Variables	Beta Coefficient (95%CI)	*p*-Value	Beta Coefficient (95%CI)	*p*-Value
Age	0.24 (0.200, 0.272)	<0.001	0.19 (0.164, 0.286)	<0.001
Sex	0.33 (0.258, 0.393)	<0.001	0.14 (0.098, 0.179)	<0.001
Race/Ethnicity	0.09 (0.044, 0.134)	<0.001	0.020 (−0.004, 0.045)	0.108
Co-occurring depression with respiratory conditions	−0.002 (−0.078, 0.073)	0.095	0.10 (0.054, 0.145)	<0.001
Insurance	−0.07 (−0.125, −0.008)	<0.05	−0.06 (−0.094, −0.026)	<0.001
Chronic conditions	0.31 (0.257, 0.369)	<0.001	0.29 (0.250, 0.325)	<0.001
Region	−0.010 (−0.022, 0.042)	0.543	−0.02 (−0.037, −0.003)	<0.05

^a^ Log transformed. CI: Confidence Interval.

## Data Availability

MEPS data are available in a publicly accessible repository through Agency for Healthcare Research and Quality (AHRQ) and the National Center for Health Statistics (NCHS). The data presented in this study are openly available on the Medical Expenditure Panel Survey (MEPS) website at https://www.meps.ahrq.gov/mepsweb/ (accessed on 21 July 2020).

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
