# Peer review of "Co-Occurring Depression and Associated Healthcare Utilization and Expenditure in Individuals with Respiratory Condition: A Population-Based Study"

_pharmacy, 2021, doi:10.3390/pharmacy9040157_

Round 1

Reviewer 1 Report

Thank you for an opportunity to review your valuable research. This study shows that people who had depression with respiratory conditions are associated with increased healthcare utilization and expenditures. I also believe that if patients who had respiratory condition with depression improve their health condition, it might be reduce healthcare utilization and expenditures.

This is a very interesting paper, using adequate index, well written, with an appropriate statistical analysis. However, some statistical analysis is ambiguous in the methods and results section. Major or minor comments are detailed below.

In abstract section (page 1, line 19), please present which regression you used.

In introduction section (page 2, line 46-49), there was no reference for the relationship between asthma and depression according to financial and physical hardships, and the sentences in terms of the relationship between antidepressants and quality of life (line 52-57) were also no reference. The authors have to mention based on the previous studies. Please add up the reference.

In Materials and Methods section (page 5, line 203-205), this study was used the regression model but there was no information which type of regression was used. Please explain which specific regression model was used; for example, ‘this study was used multiple linear regression model adjusted for age,………’ . Additionally, this study was used panel database, the authors collected their population from 2011 to 2017. I wonder that the authors used repeated measure or not. This is important point because the statistical analysis may change and the result might also be changed whether population were repeated or not. Please clarify it.

Moreover, how did the authors treat the missing data? The authors should provide this information.

In results section (page 7, line 288-291), the authors give to readers which covariates were adjusted in the below the table as foot notes, also please provide the results of full model in table 4 presenting in the supplementary file.

Author Response

We would like to thank reviewer for his/her valuable suggestions and comments. It has certainly helped us to improve our manuscript. We hope that our revision satisfies reviewers’ concerns and comments. Appended below is the point wise response to reviewers’ comments in italics.

Reviewer 1

Thank you for an opportunity to review your valuable research. This study shows that people who had depression with respiratory conditions are associated with increased healthcare utilization and expenditures. I also believe that if patients who had respiratory condition with depression improve their health condition, it might be reduce healthcare utilization and expenditures.

This is a very interesting paper, using adequate index, well written, with an appropriate statistical analysis. However, some statistical analysis is ambiguous in the methods and results section. Major or minor comments are detailed below.

We sincerely thank reviewer for his/her positive and encouraging words and finding our research valuable.

In abstract section (page 1, line 19), please present which regression you used.

We thank reviewer for pointing out this obvious omission. We have corrected this omission and the sentence now reads as “multiple linear regression analysis was performed………………..”. 

In introduction section (page 2, line 46-49), there was no reference for the relationship between asthma and depression according to financial and physical hardships, and the sentences in terms of the relationship between antidepressants and quality of life (line 52-57) were also no reference. The authors have to mention based on the previous studies. Please add up the reference.

We thank reviewer for pointing this out. We have now included the appropriate references in support of these statements.  

In Materials and Methods section (page 5, line 203-205), this study was used the regression model but there was no information which type of regression was used. Please explain which specific regression model was used; for example, ‘this study was used multiple linear regression model adjusted for age,………’ . Additionally, this study was used panel database, the authors collected their population from 2011 to 2017. I wonder that the authors used repeated measure or not. This is important point because the statistical analysis may change and the result might also be changed whether population were repeated or not. Please clarify it.

We thank reviewer for pointing out this omission. As stated above, we have now included the regression model that was used. The modified sentence now reads as “multiple linear regression analysis was performed to identify the …………………………”. 

We agree with the reviewer’s comment about use of panel survey data. We used multiple years of data to achieve an adequate sample size.The MEPS survey panel design includes 5 rounds of interviews covering 2 full calendar years of data on survey participants. Weconsolidated and used data for year 2011 to 2017 to identify demographic information, healthcare utilization and expenditures related to conditions of interest for our analyses. We did not use repeated measures since we were only interested in estimating average total healthcare utilization and expenditures between depressed and non-depressed individuals with respiratory condition using this consolidated data rather than looking at the annual healthcare utilization and expenditures or performing any trend analyses.        

Moreover, how did the authors treat the missing data? The authors should provide this information.

We thank reviewer for pointing this out. We used imputation approach by adding an extra category for the variable indicating missingness for categorical variables and treated missingness at random for continuous variables. This is now clarified in the methods section.  

In results section (page 7, line 288-291), the authors give to readers which covariates were adjusted in the below the table as foot notes, also please provide the results of full model in table 4 presenting in the supplementary file.

We sincerely apologize for this ambiguity in our description. The Table 4 represents the full model with all seven covariates included. These covariates were chosen based on the prior evidence from the literature. Five of them were found to be a significant predictor for healthcare utilization, whereas six for healthcare expenditure out of the seven covariates.   

Reviewer 2 Report

Review points

  1. Line 149 and 150.

 These conditions were categorized into three groups

It seems the sentence has ended without providing the 3 groups.

  1. Line 210 : 1. Prevlence of Depression

Correction of spelling

  1. Line 210 : Table 1 presents the demographic characteristics of the depressed and non-
    depressed individuals with respiratory conditions.

Suggestion: This sentence may be removed, only mention the table at the end of the paragraph.

In the results, few sentences are necessary details are in the relevant table.

Same for other tables.

  1. References style

Suggestion: Number of authors before  et al. should be uniform

Author Response

We would like to thank reviewer for his/her valuable suggestions and comments. It has certainly helped us to improve our manuscript. We hope that our revision satisfies reviewers’ concerns and comments. Appended below is the point wise response to reviewers’ comments in italics.

Review points

  1. Line 149 and 150.

 These conditions were categorized into three groups

It seems the sentence has ended without providing the 3 groups.

We sincerely regret this omission. We have corrected it. The modified sentence now reads as “were categorized into three groups as ≤2, 3-5 and >5 chronic disease conditions.

  1. Line 210 : 1. Prevlence of Depression

Correction of spelling

 We thank reviewer for pointing this out. This spelling is corrected now.

  1. Line 210 : Table 1 presents the demographic characteristics of the depressed and non- 
    depressed individuals with respiratory conditions.

Suggestion: This sentence may be removed, only mention the table at the end of the paragraph.

In the results, few sentences are necessary details are in the relevant table.

Same for other tables.

We thank reviewer for pointing this out, these sentences at the beginning of the paragraphs are now removed and table numbers in parentheses were inserted at the end of the paragraphs.

  1. References style

Suggestion: Number of authors before  et al. should be uniform

We thank reviewer for pointing this out. We have corrected references to keep them consistent in style.

Reviewer 3 Report

The analysis in the manuscript regarded the entire US as a population, which could be risky for two reasons.

  1. The data used was collected across the US, while different states naturally have different demographic characteristics and different medical treatment costs.
  2. Health conditions such as asthma, emphysema, chronic bronchitis, and depression could likely be different naturally for different states, depending on the state's development level, economical activities, environmental conditions, demographic characteristics, etc. 

The manuscript could be improved if the population is reduced to a certain state or a certain healthcare system.

Author Response

We would like to thank reviewer for his/her valuable suggestions and comments. It has certainly helped us to improve our manuscript. We hope that our revision satisfies reviewers’ concerns and comments. Appended below is the point wise response to reviewers’ comments in italics.

The analysis in the manuscript regarded the entire US as a population, which could be risky for two reasons.

  1. The data used was collected across the US, while different states naturally have different demographic characteristics and different medical treatment costs.

We agree with the reviewer’s comment that we have used national rather than regional data. The aim of our investigation was to explore the national level prevalence, pattern of healthcare use and expenditures in non-institutionalized US population having co-occurring depression with respiratory condition. We chose the Medical Expenditure Panel Survey (MEPS) which represents a nationwide survey of the US population and their healthcare providers to accomplish this goal. To analyze the impact of different demographic characteristics such as age, sex, race/ethnicity, co-occurring depression, health insurance, comorbidity, and different region on healthcare utilization and expenditure, we performed the multiple linear regression analyses. The MEPS was not primarily designed to facilitate state or local-level estimation; the smallest geographic region available in the MEPS data files is census region (Northeast, Midwest, South, and West) which we have used in our analyses.

  1. Health conditions such as asthma, emphysema, chronic bronchitis, and depression could likely be different naturally for different states, depending on the state's development level, economical activities, environmental conditions, demographic characteristics, etc. 

We agree with the reviewer’s comment that health conditions such as asthma, emphysema, chronic bronchitis, and depression could likely be different naturally for different states, depending on the state's development level, economical activities, environmental conditions, demographic characteristics, etc. As stated above, the goal of our investigation was to explore the national level prevalence, pattern of healthcare use and expenditures in non-institutionalized US population having co-occurring depression with respiratory condition while adjusting for demographic characteristics such as age, sex, race/ethnicity, co-occurring depression, health insurance, comorbidity, and different region. Collecting data on state’s developmental level, economic activities, and environmental conditions is not a goal of this survey and hence such data is not a part of this database that can be analyzed. This survey is specifically designed to capture nationally representative estimates of health care use, expenditures, and insurance coverage for the civilian non-institutionalized population.

The manuscript could be improved if the population is reduced to a certain state or a certain healthcare system.

We thank reviewer for the suggestion. Although, limiting investigation to certain state or healthcare system will reduce the required sample size that is needed to achieve the necessary power for the study to detect meaningful differences if they exist and to carry out such investigation.We used multiple years of national survey data to achieve an adequate sample size for this current investigation. Moreover, the primary goal of our investigation was to explore the national level prevalence, pattern of healthcare use and expenditures in non-institutionalized US population having co-occurring depression with respiratory condition. As stated above, the MEPS was not primarily designed to facilitate state or local-level estimation; the smallest geographic region available in the MEPS-HC data files is census region (Northeast, Midwest, South, and West) which we have used in our analyses.